# Meiotic drive of female-inherited supernumerary chromosomes in a pathogenic fungus

Michael Habig[1,2], Gert HJ Kema[3,4], Eva Holtgrewe Stukenbrock[1,2]*

[1]Environmental Genomics, Christian-Albrechts University of Kiel, Kiel, Germany; [2]Max Planck Institute for Evolutionary Biology, Plön, Germany; [3]Wageningen Plant Research, Wageningen University and Research, Wageningen, The Netherlands; [4]Laboratory of Phytopathology, Wageningen University and Research, Wageningen, The Netherlands

**Abstract** Meiosis is a key cellular process of sexual reproduction that includes pairing of homologous sequences. In many species however, meiosis can also involve the segregation of supernumerary chromosomes, which can lack a homolog. How these unpaired chromosomes undergo meiosis is largely unknown. In this study we investigated chromosome segregation during meiosis in the haploid fungus *Zymoseptoria tritici* that possesses a large complement of supernumerary chromosomes. We used isogenic whole chromosome deletion strains to compare meiotic transmission of chromosomes when paired and unpaired. Unpaired chromosomes inherited from the male parent as well as paired supernumerary chromosomes in general showed Mendelian inheritance. In contrast, unpaired chromosomes inherited from the female parent showed non-Mendelian inheritance but were amplified and transmitted to all meiotic products. We concluded that the supernumerary chromosomes of *Z. tritici* show a meiotic drive and propose an additional feedback mechanism during meiosis, which initiates amplification of unpaired female-inherited chromosomes.

DOI: https://doi.org/10.7554/eLife.40251.001

*For correspondence:
stukenbrock@evolbio.mpg.de

Competing interests: The authors declare that no competing interests exist.

## Introduction

In eukaryotes, meiosis is a highly conserved mechanism that generates gametes and includes recombination and pairing of homologous chromosomes. Meiosis combines one round of DNA replication with two subsequent rounds of chromosome segregation (reviewed in (*Klutstein and Cooper, 2014*; *Zickler and Kleckner, 2015*)). DNA replication during the meiotic S-phase progression is coupled directly to interactions between homologous sequences and results in the pairing of chromosomes and recombination (*Cha et al., 2000*). The initial pairing of homologous chromosomes is important for meiosis and proper chromosome segregation (reviewed in (*Loidl, 2016*)). However, it is less clear how meiosis proceeds when pairing of homologous chromosomes does not take place due to unequal sets of chromosomes, as is the case in organisms with non-essential supernumerary chromosomes.

Supernumerary chromosomes, also known as B chromosomes, conditionally dispensable chromosomes or accessory chromosomes, are present in some but not all members of a population. Supernumerary chromosomes have been described in a large variety of plant, animal and fungal species, including for example in 14% of karyotyped orthopteran insect species (*Jones, 1995*), 8% of monocots, and 3% of eudicot species (*Levin et al., 2005*). These chromosomes commonly show non-Mendelian modes of inheritance, leading to segregation distortion during meiosis and a change in the frequency of the supernumerary chromosome in the progeny – a process that has been described as

chromosome drive (*Jones et al., 2008*; *Valente et al., 2017*). Segregation advantage of supernumerary chromosomes can be due to drive mechanisms at the pre-meiotic, meiotic, or post-meiotic stages of gamete formation (*Hasegawa, 1934*; *Houben et al., 2014*; *Houben, 2017*; *Mroczek et al., 2006*; *Ohta, 1996*) and has been demonstrated in animals and plants (*Akera et al., 2017*; *Mroczek et al., 2006*). In fungi, supernumerary chromosomes have been characterized in several species and notably studied in fungal pathogens where their presence in some cases is associated with virulence (*Ma et al., 2010*; *Miao et al., 1991*). The underlying mechanisms causing non-Mendelian inheritance of the supernumerary chromosomes in fungi are however poorly understood.

The genomic composition of the fungal plant pathogen *Zymoseptoria tritici* provides an attractive model to analyze supernumerary chromosome transmission. The genome of this fungus contains one of the largest complements of supernumerary chromosomes reported to date (*Goodwin et al., 2011*). The eight distinct supernumerary chromosomes (chr14 to chr21) of the reference isolate IPO323 show presence/absence polymorphisms among isolates and differ in their genetic composition compared to the essential chromosomes (*Goodwin et al., 2011*; *Plissonneau et al., 2016*). The supernumerary chromosomes in *Z. tritici* are enriched in repetitive elements (*Dhillon et al., 2014*; *Goodwin et al., 2011*; *Grandaubert et al., 2015*), mainly heterochromatic (*Schotanus et al., 2015*) and frequently lost during mitosis (*Möller et al., 2018*) and meiosis (*Croll et al., 2013*; *Fouché et al., 2018*; *Goodwin et al., 2011*; *Wittenberg et al., 2009*) and they show a considerably lower recombination rate compared to the core chromosomes (*Croll et al., 2015*; *Stukenbrock and Dutheil, 2018*). However, core and supernumerary chromosomes share many repetitive element families and their subtelomeric regions contain the same transposable element families (*Dhillon et al., 2014*; *Grandaubert et al., 2015*; *Schotanus et al., 2015*). In contrast to many gene-poor supernumerary chromosomes described in plants and animals, those in *Z. tritici* possess a relatively high number of protein-coding genes (727, corresponding to 6% of all genes) (*Grandaubert et al., 2015*). Recently, we demonstrated that the supernumerary chromosomes of *Z. tritici* confer a fitness cost: Isogenic strains lacking distinct supernumerary chromosomes produce higher amounts of asexual spores during host infection when compared to wild type with the complete set of supernumerary chromosomes (*Habig et al., 2017*). Despite the instability and fitness cost of the supernumerary chromosomes, they have been maintained over long evolutionary periods (*Stukenbrock et al., 2011*; *Stukenbrock and Dutheil, 2018*), and it is therefore intriguing to address the mechanisms of supernumerary chromosome maintenance in the genome of *Z. tritici*.

Here, we used *Z. tritici* with its unique set of supernumerary chromosomes as a model to study the dynamics of unpaired chromosomes during meiosis. *Z. tritici* is a heterothallic, haploid ascomycete (i.e. two individuals of different mating type [*mat1-1* and *mat1-2*] are required to form a diploid zygote) (*Kema et al., 1996*; *Kema et al., 2018*). If two haploid cells of opposite mating types contain a different complement of supernumerary chromosomes, the resulting diploid zygote consequently contains unpaired chromosomes. Upon Mendelian segregation during meiosis (segregation of the homologous chromosomes during meiosis I followed by chromatid segregation during meiosis II), 50% of the resulting ascospores are predicted to contain the unpaired chromosomes (*Figure 1A*). To test this prediction, we performed crosses between isolates with different subsets of supernumerary chromosomes (*Figure 1B*). Based on controlled experiments and tetrad analyses, we surprisingly found that the supernumerary chromosomes of *Z. tritici* are subject to a meiotic drive restricted to unpaired chromosomes inherited from the female parent. Our results suggest that this drive mechanism is due to an additional, female-specific amplification of unpaired chromosomes during meiosis, a process that can ensure the maintenance of these chromosomes over long evolutionary periods.

## Results

### Unpaired supernumerary chromosomes show drive correlated with mitochondrial transmission

To test the transmission of supernumerary chromosomes during meiosis we used the reference strain IPO323 (mating type *mat1-1*) and eight isogenic chromosome deletion strains (IPO323Δchr14-21, mating type *mat1-1*) generated in a previous study (*Habig et al., 2017*). Each of the chromosome deletion strains differs in the absence of one supernumerary chromosome, thereby allowing us to compare the transmission of individual chromosomes in a paired and an unpaired state. We crossed

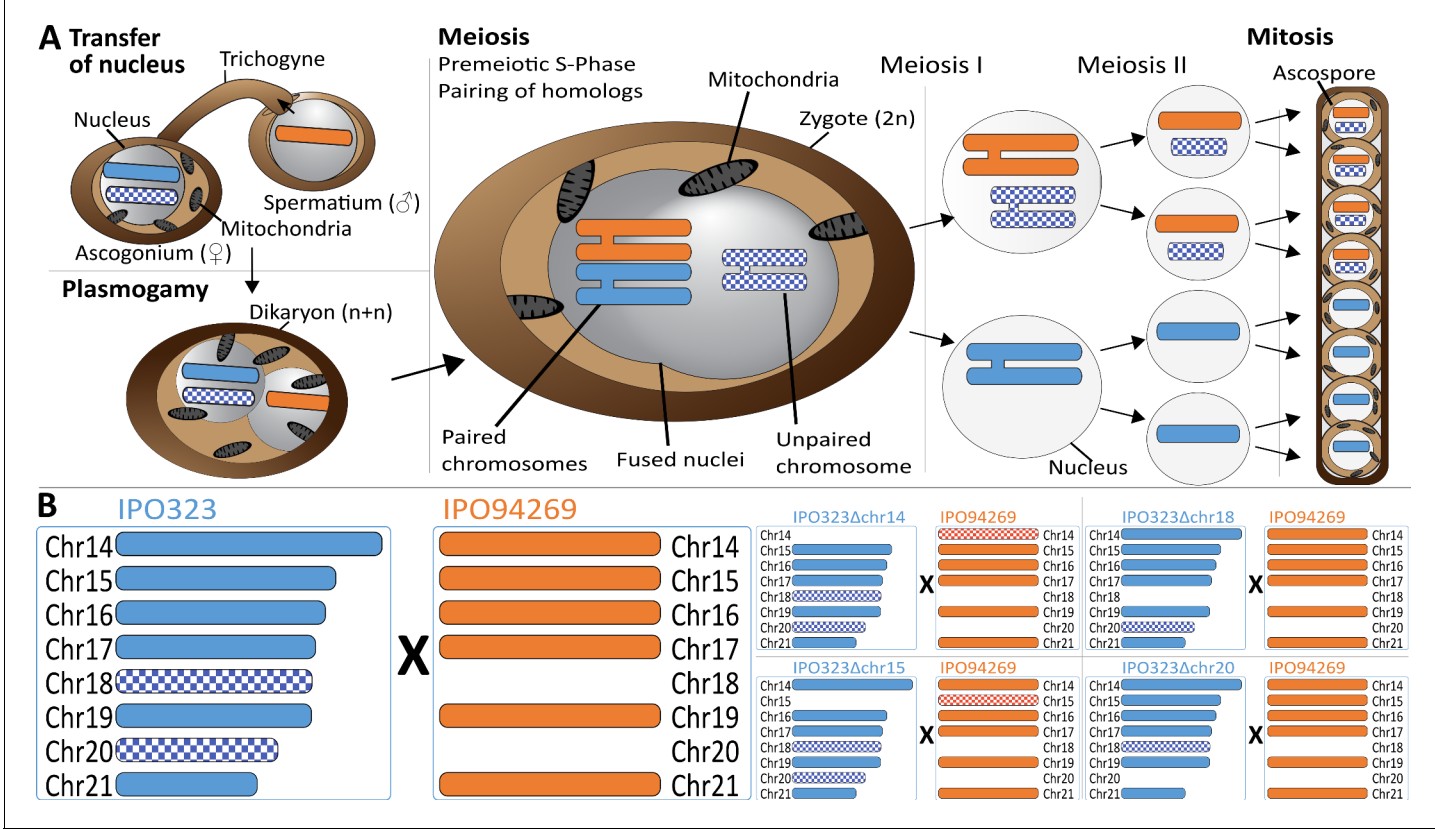

**Figure 1.** Meiosis and chromosome segregation in *Z.tritici*. (**A**) Schematic overview of the assumed sexual process between two parental strains of *Z. tritici* (*Alexopoulos et al., 1996*; *Kema et al., 2018*) with one supernumerary chromosome shared and therefore paired (blue/orange) and one supernumerary chromosome unique to one strain (blue checkered) and unpaired in the zygote. The spermatial nucleus is transferred from the male partner via the trichogyne to the ascogonium of the female partner, resulting in plasmogamy and a dikaryon with two separate nuclei. Prior to karyogamy, the chromosomes are replicated and thus comprise each two chromatids when meiosis is initiated by pairing of homologous chromosomes. In meiosis I, homologous chromosomes are segregated, followed by chromatid separation in meiosis II. A subsequent mitosis results in the production of eight ascospores contained within one ascus. The expected segregation of chromosomes according to Mendelian law of segregation is shown - which for unpaired chromosomes is 4:0. (**B**) Schematic illustration of the distribution of supernumerary chromosomes present in the parental strains exemplified for five of nine different crosses performed in this study. Parental strain IPO323 contains eight supernumerary chromosomes (chr14-21, blue). Parental strain IPO94269 contains six supernumerary chromosomes with homologs in IPO323 (chr14, chr15, chr16, chr17, chr19, and chr21 in orange). The IPO323 chromosomes chr18 and chr20 are not present in IPO94269. We used a set of IPO323 chromosome deletion strains to generate an additional unpaired chromosome (as examples IPO323Δchr14 X IPO94269, IPO323Δchr15 X IPO94269, IPO323Δchr18 X IPO94269, IPO323Δchr20 X IPO94269 to demonstrate the one to three unpaired chromosomes and the five to six paired chromosomes present in the different crosses). Orange and blue indicate chromosomes that are shared between both strains. Checkered orange and checkered blue indicate chromosomes that are unique to one parent and therefore unpaired in the zygote.

DOI: https://doi.org/10.7554/eLife.40251.002

The following figure supplement is available for figure 1:

**Figure supplement 1.** Synteny comparison of IPO94269 and IPO323.

DOI: https://doi.org/10.7554/eLife.40251.003

these strains, *in planta*, with another *Z. tritici* isolate: IPO94269 (*mat1-2*) (*Figure 1B*) in three separate experiments (A, B and C) and used a combination of PCR assays, electrophoretic karyotyping and whole genome sequencing to assess the segregation of chromosomes during meiosis. IPO94269 contains six supernumerary chromosomes homologous to the IPO323 chromosomes 14, 15, 16, 17, 19, and 21 (*Figure 1—figure supplement 1*) (*Goodwin et al., 2011*). The experiments included a total of 39 crosses of IPO323/IPO323 chromosome deletion strains with IPO94269 resulting in different complements of paired and unpaired supernumerary chromosomes in the diploid zygote (Table 1, *Supplementary file 1*). We hypothesized that the inheritance of the unpaired supernumerary chromosomes could be linked to the female or male role of the parental strain. Sexual

mating of heterothallic fungi of the genus *Zymoseptoria* involves a female partner that produces a sexual structure called the ascogonium. The ascogonium receives the spermatium with the male nucleus from the fertilizing male partner through a particular structure called the trichogyne (*Crous, 1998*) (*Figure 1A*). Importantly, the same strain can act as either the female or male partner (*Kema et al., 2018*). Mitochondrial transmission is generally associated with the female structure (*Ni et al., 2011*). We used specific mitochondrial PCR based markers to distinguish the mitochondrial genotype in the progeny and thereby determine which of the two parental strains (in this case IPO323 or IPO94269) acted as a female partner in a cross.

In all three experiments the ascospore progeny showed either the mitochondrial genotype of IPO94269 or IPO323. Therefore, both strains can act as the female and male partner during crosses (*Supplementary file 2–4*). However, transmission of the mitochondrial genotype varied significantly between experiments with the relative frequency of the IPO94269 mitochondrial genotype in the progeny being 80%, 11% and 65% in experiment A, B and C, respectively (*Figure 2A*). Interestingly, the transmission of unpaired chromosomes correlated to the sexual role (female/male) of the parent from which the unpaired chromosome was inherited. Unpaired chromosomes inherited from IPO94269 were underrepresented among ascospores with the IPO323 parent mitochondrial genotype (*Figure 2B*). In contrast, the unpaired supernumerary chromosomes 18 and 20, which were always inherited from the parent IPO323, were highly overrepresented among ascospores with the IPO323 mitochondrial genotype (*Figure 2B*). For ascospores with the mitochondrial type of the IPO94269 parent, this segregation distortion was reversed. Unpaired chromosomes inherited from IPO94269 (with the exception of unpaired chr14) were highly overrepresented among ascospores with the mitochondrial genotype of the IPO94269 parent. On the other hand, the unpaired supernumerary chromosomes 18 and 20, always inherited from the IPO323 parent, were underrepresented among ascospores with the mitochondrial genotype of the IPO94269 parent (*Figure 2B*).

Although the transmission of the supernumerary chromosomes was highly similar between experiments A, B and C when the mitochondrial genotype was used to group the data (*Figure 2—figure supplement 1A*), the overall transmission of the supernumerary chromosomes varied considerably between the experiments due to the highly divergent mitochondrial genotype inheritance in the three experiments (*Figure 2—figure supplement 1B*). However, we find a clear transmission advantage for all supernumerary chromosomes, except chromosome 14, when pooling all data from the three experiments (transmission to more than 50% of the progeny) (*Figure 2—figure supplement 1C*). Based on these observations, we conclude that unpaired supernumerary chromosomes show a chromosome drive mechanism, but this drive is restricted to chromosomes inherited from the mitochondria-donating female parent.

## Transmission of mitochondria is affected by the cell density

We next asked which factors determine the sexual role and thereby the mitochondrial inheritance in the sexual crosses of *Z. tritici*. Recently, competition between sexual and asexual modes of reproduction in *Z. tritici* was shown to be affected by the cell density (*Suffert et al., 2018*). We therefore hypothesized that the cell density of the two parental strains could also affect the sexual roles of the two parental strains. To test this, we set up crosses between the strains IPO323Δchr19 and IPO94269 in which the cell density varied from $10^5$ cells to $10^7$ cells/mL of each of the parental strains. To distinguish the female and male partner in the crosses we again assessed the mitochondrial transmission frequencies. Interestingly, we find that the cell density of the two parental strains strongly correlates with the transmission of the mitochondrial genotype. Crosses with a lower cell density of IPO323Δchr19 resulted in a higher proportion of the progeny carrying the IPO323 mitochondrial genotype (*Figure 2C*). Similarly, a lower cell density of IPO94269 resulted in a higher proportion of the progeny carrying the IPO94269 mitochondrial genotype. This illustrates that a density-dependent mechanism affects the sexual role of the *Z. tritici* strains during sexual mating.

Variation in the sexual role in turn affected the transmission of unpaired supernumerary chromosomes. The unpaired chromosome 19 inherited from the parent IPO94269 increased in frequency in the meiotic progeny with increasing frequency of the IPO94269 mitochondrial genotype. Unpaired chromosome 18 and chromosome 20 inherited from the parent IPO323Δchr19 increased in frequency in the meiotic progeny with an increase in frequency of the IPO323 mitochondrial genotype (*Figure 2D*). Therefore, a clear effect of cell density is discernable and we conclude that environmental factors that affect the infection density of different *Z. tritici* strains also strongly affect the sexual

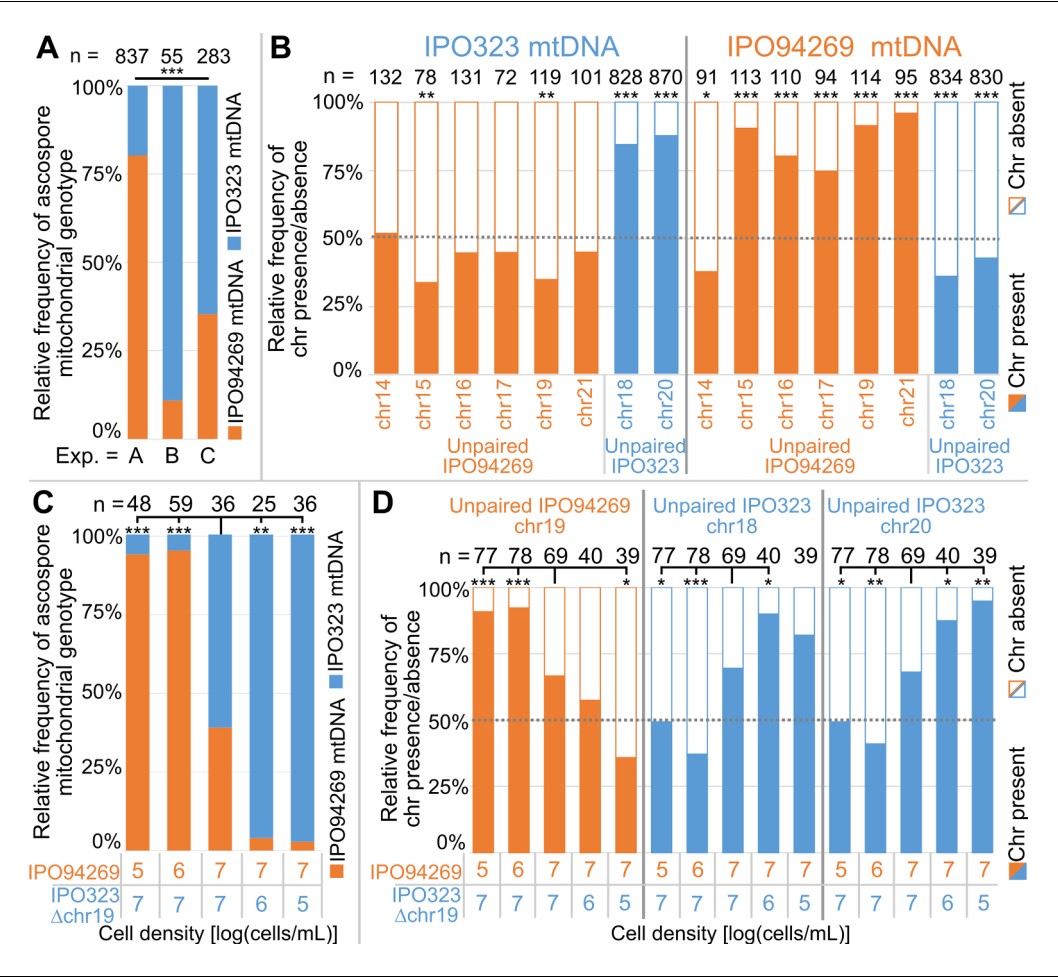

**Figure 2.** Unpaired supernumerary chromosomes show a segregation advantage only when inherited from the female parent. (**A**) Relative frequencies of mitochondrial genotypes in random and randomized ascospores in experiments A, B and C. The mitochondrial transmission varied significantly between the three experiments. Statistical significance was inferred by Fisher's exact test ($p < 2.2*10^{-16}$). (**B**) Relative frequencies of the presence and absence of unpaired supernumerary chromosomes in all progeny ascospores pooled for experiments A, B and C according to the mitochondrial genotype of the ascospore. Orange and blue indicate unpaired chromosomes originating from IPO94269 or IPO323, respectively. Unpaired chromosomes (with the exception of chr14) inherited from the parent that provided the mitochondrial genotype (i.e. the female parent) are overrepresented in the progeny, while the same chromosomes when originating from the male parent are not. Statistical significance was inferred by a two-sided binomial test with a probability of $p=0.5$. (**C**) Cell density affects the sexual role during mating and thereby the transmission of the mitochondria. Relative frequencies of mitochondrial genotype in random and randomized ascospores isolated from crosses of IPO94269 and IPO323Δchr19 that were co-inoculated on wheat at different cell densities. The resulting progeny shows a correlation between cell-density and mitochondria transmission. Strains inoculated at lower density in general take the female role as observed by the mitochondrial transmission. Statistical significance was inferred by a two-sided Fisher's exact test compared to the co-inoculation with equal cell densities of both strains. (**D**) The cell density affects the transmission of unpaired chromosomes. Relative frequencies in all ascospores of the presence and absence of unpaired supernumerary chromosomes 19, inherited from parent IPO94269 and unpaired chromosomes 18 and 20, inherited from IPO323 Δchr19 are indicated according to the cell density of the parental strains IPO94269 and IPO323Δchr19 at inoculation. Statistical significance was inferred by a two-sided Fisher's exact test compared to the co-inoculation with equal cell density of both strains. (*=$p < 0.05$, **=$p < 0.005$, ***=$p < 0.0005$, see *Supplementary file 5* for details on all statistical tests).

DOI: https://doi.org/10.7554/eLife.40251.004

The following figure supplement is available for figure 2:

**Figure supplement 1.** Transmission of unpaired chromosomes and mitochondria.

*Figure 2 continued on next page*

*Figure 2 continued*

DOI: https://doi.org/10.7554/eLife.40251.005

role of strains and thereby the transmission of supernumerary chromosomes. We hypothesize that variability in environmental factors between the three experiments, which were conducted during different seasons and at different locations, may explain the observed variability in the frequency of sexual roles of the two strains among the three experiments. Accordingly, the transmission of unpaired supernumerary chromosomes may also be affected by seasonal changes in environmental factors.

## Paired supernumerary chromosomes show Mendelian segregation with frequent losses

In *Z. tritici*, as in other ascomycetes, one meiosis produces eight ascospores by an additional mitosis following meiosis (*Ni et al., 2011*; *Wittenberg et al., 2009*). The outcome of single meiotic events can be analyzed by tetrad analyses whereby the eight ascospores of a tetrad - in ascomycetes the ascospores contained in an ascus - are isolated and genotyped. We used tetrad analyses to address how paired supernumerary chromosomes segregate during meiosis. For a total of 24 separate asci, we verified that all eight ascospores originated from the same ascus and were the products of a single meiosis using six segregating markers located on the essential chromosomes (Table 2). With these 24 asci we could identify segregation patterns and furthermore eliminate post-meiotic effects on the observed chromosomal frequencies. Each tetrad allowed the analysis of the segregation pattern for both unpaired and paired chromosomes. First, we focused on the segregation of paired chromosomes within these tetrads. In the 24 asci we could observe the transmission of paired supernumerary chromosomes in 129 instances. We could discern the segregation of each paired supernumerary chromosome using specific segregating markers for each of the supernumerary chromosomes from both parental strains. In general, the paired supernumerary chromosomes showed a Mendelian segregation pattern (*Figure 3*). Of the 129 instances of supernumerary chromosome pairing, 120 (93%) showed a Mendelian segregation pattern with the expected 4:4 ratio (*Figure 3*, black outline) and only nine instances (7%) showed a non-Mendelian transmission pattern deviating from the 4:4 ratio (*Figure 3*, red outline). In no instances did the number of ascospores with a segregating marker for a paired supernumerary chromosome exceed the four ascospores predicted by Mendelian segregation. In two of the nine instances however, we found ascospores with two copies of supernumerary chromosome 21, representing one copy from IPO323 and another from IPO94269. Whole genome sequencing validated the presence of two copies of chromosome 21 in the genomes of these ascospores. In the two additional ascospores of the tetrad analysis the chromosome 21 was missing (*Figure 3—figure supplement 1A–C*) suggesting that the non-Mendelian transmission patterns are due to loss of chromosomes, non-disjunction of sister chromatids, or non-disjunction of homologous chromosomes during meiosis (*Wittenberg et al., 2009*).

Whole genome sequencing of two tetrads allowed for the dissection of the inheritance of SNPs on paired supernumerary and core chromosomes. For both paired supernumerary and core chromosomes stretches of haplotypes of parental IPO94269 SNPs can be recognized in the ascospores as a result of recombination and crossover events (*Figure 3—figure supplement 2A–F*). These IPO94269 haplotypes are consistently present in four of the eight ascospores indicating a Mendelian segregation. A more detailed analyses of the SNP distribution showed that the majority of individual IPO94269 SNPs are present in four of the eight ascospores as expected (113135 of a total 167346 SNPs in ascus A03-4 and 115612 of a total 170537 SNPs in ascus A08-1) (*Figure 3—figure supplement 2C and F*, *Supplementary file 5*). However a number of SNPs was found to be present only in three (10.5% and 10.7% of all SNPs in ascus A03-4 and A08-1, respectively), two (8.9% and 9.0% of all SNPs) or one ascospores (12.7% and 12.8% of all SNPs) of a tetrad, which could indicative non-Mendelian transmission for a subset of SNPs located on the core and supernumerary chromosomes. To further investigate the occurrence of non-Mendelian SNPs we restricted our analyses to include only SNPs in regions of the genome alignment with high read coverage. Restricting the analysis to SNPs with >8X read coverage in at least one ascospore substantially increased the relative frequency of SNPs detected in exactly four ascospores to 94% (96498 of a total 102797 SNPs in ascus A03-4

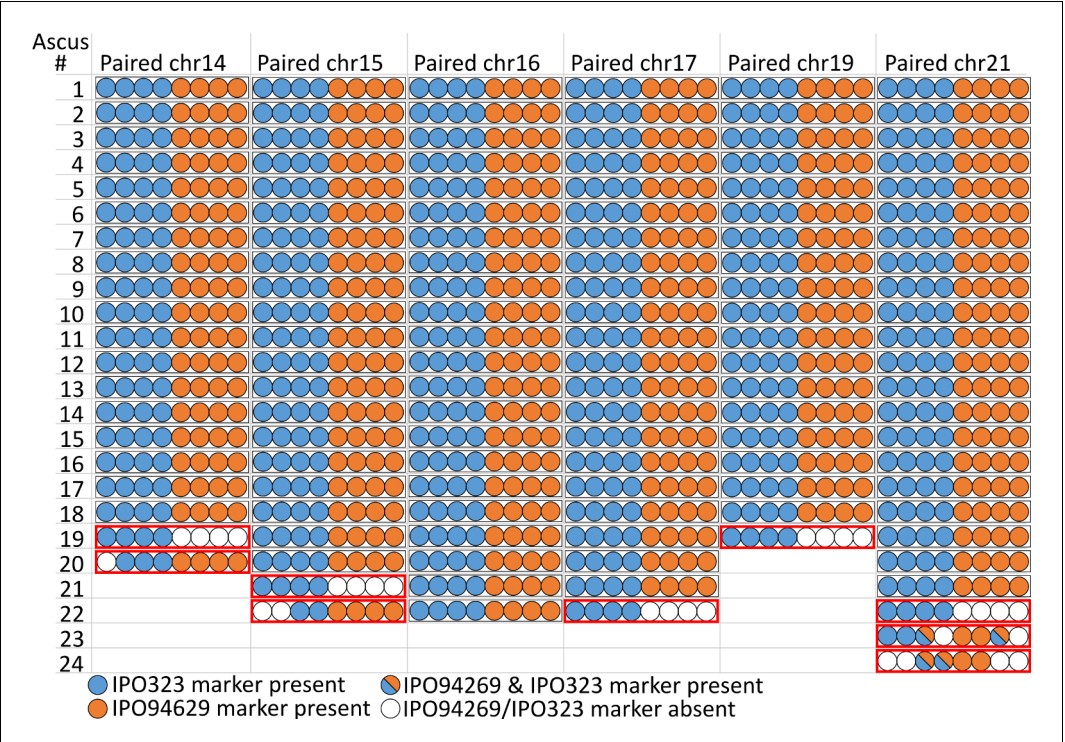

**Figure 3.** Paired supernumerary chromosomes show Mendelian segregation with frequent losses. Analysis of segregation of paired supernumerary chromosomes in 24 complete tetrads. The transmission of chromosomes 14, 15, 16, 17, 19, and 21 with homologs in both parental strains IPO94269 and IPO323 was detected using segregating markers for chromosomes inherited from the parental strains IPO94269 (orange) and IPO323/IPO323Δchr14-21 (blue) in the eight ascospores originating from 24 asci. In 120 of the 129 cases (black outline) we observed a 4:4 ratio in the progeny. Note: for crosses/chromosome combinations where no paired chromosome was present no ascus is shown, which reduces the number of shown asci from the 24 asci that were analyzed in both *Figure 3* and *Figure 4*.

DOI: https://doi.org/10.7554/eLife.40251.006

The following source data and figure supplements are available for figure 3:

**Source data 1.** Examples of gel electrophoresis of PCR products for (**A**) core chromosome 13 marker on mating type, (**B**) core chromosome 4 marker 11O21, (**C**) core chromosome 4 marker 04L20, (**D**) core chromosome three marker caa-0002, core chromosome five marker ggc-001, core chromosome seven marker ac-001, (**E**) segregating marker of the mitochondrial genotype, (**F**) segregating marker of the supernumerary chromosome 14, (**G**) segregating marker of the supernumerary chromosome 15, (**H**) segregating marker of the supernumerary chromosome 16, (**I**) segregating marker of the supernumerary chromosome 17, (**J**) segregating marker of the supernumerary chromosome 19, (**K**) segregating marker of the supernumerary chromosome 19, (**L**) subtelomeric marker of the supernumerary chromosome 19, (**M**) subtelomeric marker of the supernumerary chromosome 19, (**N**) segregating marker of the supernumerary chromosome 21, (**O**) subtelomeric marker of the supernumerary chromosome 18, (**P**) centromeric marker of the supernumerary chromosome 18, (**Q**) subtelomeric marker of the supernumerary chromosome 18, (**R**) subtelomeric marker of the supernumerary chromosome 20, (**S**) centromeric marker of the supernumerary chromosome 20, (**T**) subtelomeric marker of the supernumerary chromosome 20.

DOI: https://doi.org/10.7554/eLife.40251.009

**Figure supplement 1.** Whole genome sequencing confirms chromosome drive for unpaired supernumerary chromosomes.

DOI: https://doi.org/10.7554/eLife.40251.007

**Figure supplement 2.** Distribution of SNPs on core and supernumerary chromosomes in two fully sequenced tetrad.

DOI: https://doi.org/10.7554/eLife.40251.008

and 98954 of a total 105024 SNPs in ascus A08-1). Further increasing the fidelity of SNP detection

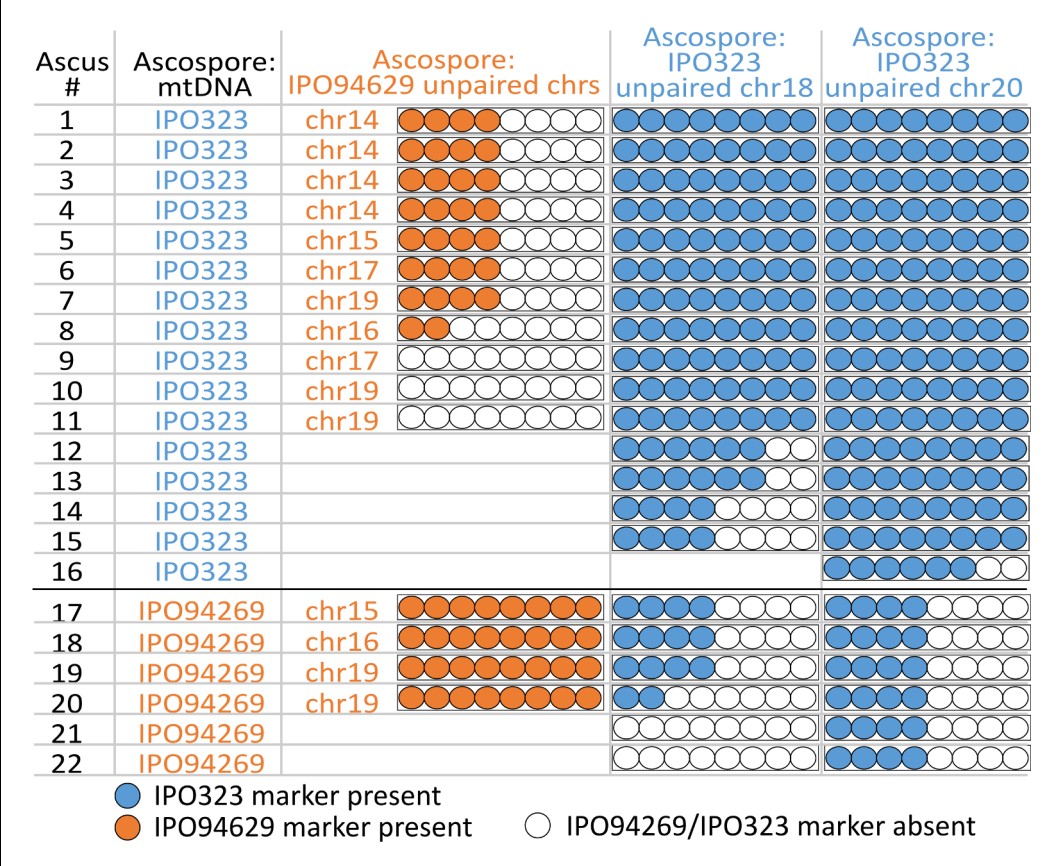

**Figure 4.** Unpaired supernumerary chromosome show meiotic drive if inherited from the female parent. Analysis of segregation of unpaired supernumerary chromosomes in 24 complete tetrads according to the mitochondrial genotype. The transmission of chromosomes unique to one of the parental strains and the mitochondrial genotype was detected using chromosomal or mitochondrial markers originating from IPO94269 (orange) and IPO323 (blue) in eight ascospores derived from 24 asci. When IPO323 was the female parent (i.e. the ascospores inherited the mitochondrial genotype of the IPO323 parent) unpaired chromosomes 18 and 20 originating from IPO323 show a strong chromosome drive and are overrepresented in the ascospores. When IPO94269 was the female parent unpaired chromosomes originating from IPO94269 show a strong chromosome drive. Unpaired chromosomes originating from the male parent (i.e. the one not donating the mitochondrial genotype) show Mendelian segregation pattern or are lost. Note: for crosses/chromosome combinations where no unpaired chromosome was present no ascus is shown, which reduces the number of shown asci from the 24 asci that were analyzed in both *Figure 3* and *Figure 4*.
DOI: https://doi.org/10.7554/eLife.40251.010

to SNPs at positions with >8X read coverage at all occurrences further increased the proportion of SNPs detected in exactly four ascospores to 97% or 98% in ascus A03-4 and A08-1, respectively (*Figure 3—figure supplement 2C and F*, *Supplementary file 5*). Very few SNPs on paired supernumerary chromosomes are present in more than four ascospores (69 and 130 SNPs in ascus A03-4 and A08-1, respectively), which could be the result of gene conversion. In conclusion, a similar pattern of the transmission of SNPs was detected for core chromosomes and paired supernumerary chromosomes consistent with Mendelian segregation for the vast majority of SNPs.

We next extended the analysis of transmission fidelity to include all ascospores isolated in experiments A, B and C to compare the rate of loss of paired supernumerary chromosomes. We assessed the rate of chromosome loss from 10078 instances of paired supernumerary chromosomes in isolated meiotic progenies. In 377 cases (3.7%) we found evidence for supernumerary chromosome loss in the ascospores based on the absence of specific chromosome markers (*Supplementary file 6*). Interestingly, the frequency of loss of paired supernumerary chromosomes varies significantly

between the individual chromosomes ($\chi^2$-Test: exp. A: p=1.96×10$^{-06}$, exp. B: p=1.72×10$^{-09}$, exp. C: p=4.18×10$^{-4}$) (*Supplementary file 6*) with chromosome 16 showing the lowest rate of loss in all three experiments. The frequency of chromosome loss, however, shows no correlation to particular chromosome characteristics like chromosome size or the extent of homology between the chromosomes from IPO323 and IPO94269 (*Figure 1—figure supplement 1B*).

## Unpaired supernumerary chromosomes inherited from the female parent show meiotic drive

Using the same 24 complete tetrads we dissected the fate of unpaired chromosomes during single meiotic events. Each tetrad contained between one to three unpaired chromosomes with chromosome 18 and 20 being solely inherited from IPO323 and chromosome 14, 15, 16, 17, and 19 being solely inherited from IPO94269 in crosses performed with the five IPO323 whole chromosome deletion strains. In contrast to paired supernumerary chromosomes, unpaired supernumerary chromosomes show distinct segregation distortion (*Figure 4*) that correlates with mitochondrial transmission; unpaired chromosomes originating from the female parent show a strong meiotic chromosome drive. On the other hand, unpaired chromosomes originating from the male parent (i.e. the parent that did not provide the mitochondria) often show a Mendelian segregation pattern and are frequently lost.

In the 24 tetrads dissected here, all eight ascospores originating from the same ascus showed the same mitochondrial genotype (*Supplementary file 3 & 4*) confirming previous results on the uniparental inheritance of mitochondria in *Z. tritici* (*Kema et al., 2018*). Isolated ascospores had the mitochondrial genotype of the parent IPO323 in 18 asci, while the ascospores of the remaining six asci showed the IPO94269 genotype, confirming that both parental strains, IPO323 and IPO94269, can act as the female parent during sexual mating with no significant difference between the two strains (two sided binomial test (p=0.5), p=0.25) (*Kema et al., 2018*). In 18 asci that showed the IPO323 mitochondrial genotype, supernumerary chromosomes 18 and 20, inherited from IPO323 (the female), were unpaired in 15 and 16 meioses, respectively (*Figure 4*). Chromosome 18 was present in all eight ascospores in 11 of the 15 asci (ascus #1–11) instead of the expected four ascospores. In two asci, the chromosome was present in six ascospores (ascus #12–13). Chromosome 20 was present in all eight ascospores (ascus #1–15) in 15 of the 16 asci and in one ascus in six ascospores (ascus #16). This transmission pattern was reversed for the six asci exhibiting the IPO94269 mitochondrial genotype (*Figure 4*). Here, the female-inherited unpaired chromosomes from IPO94269 show meiotic drive while the male-inherited unpaired chromosomes from IPO323 show Mendelian segregation pattern or were lost (*Figure 4*). We validated our PCR-karyotyping by sequencing the genomes of 16 ascospore isolates originating from two asci and mapped the resulting reads to the reference genome of IPO323. For all 16 ascospores we find similar coverage for all essential chromosomes and supernumerary chromosomes (*Figure 3—figure supplement 1*) and all 16 ascospores show similar coverage for chromosome 18 and 20 verifying the transmission of these chromosomes to the eight ascospores of each ascus instead of the expected four ascospores.

The meiotic drive of the unpaired supernumerary chromosome could imply an additional amplification step that only affects unpaired chromosomes derived from the female parent. We found however that in one cross, this additional amplification of an unpaired chromosome was incomplete: In the ascus A08-1, unpaired chromosome 18 was transmitted to all eight ascospores, but four of the ascospores contained only a partial chromosome 18, (*Figure 3—figure supplement 1F*). The partial chromosomes 18 showed a Mendelian segregation pattern, indicating that the additional amplification step of the unpaired chromosome 18 occurred prior to the first meiotic division, which in this rare case was incomplete.

## Discussion

The fate of supernumerary chromosomes during meiosis is poorly understood despite the widespread occurrence of this type of chromosome in different taxa. Here, we show that unpaired chromosomes of the fungal plant pathogen *Z. tritici* are transmitted and amplified by a meiotic drive which acts only on chromosomes inherited from the female parent (*Figure 5A*). Crossing experiments of haploid individuals of opposite mating types document this as: i) unpaired supernumerary chromosomes show chromosome drive only when inherited from the female parent; ii) unpaired

supernumerary chromosomes show a Mendelian segregation pattern and are frequently lost when inherited from the male parent; and iii) paired supernumerary chromosomes show Mendelian segregation pattern, but with frequent losses during meiosis.

Our data strongly suggest that this chromosome drive does not result from pre- or post-meiotic mechanisms but occurs during meiosis. First of all, we exclude the occurrence of a post-meiotic chromosome drive, for example killing of ascospores that did not contain the drive element, as this would make it impossible to isolate complete tetrads, which we were able to do. Second, we consider any pre-meiotic mechanisms that could affect the number of supernumerary chromosomes unlikely. These mechanisms could either be a pre-meiotic amplification (*Figure 5B*) or preferential segregation of the supernumerary chromosomes. Both mechanisms would affect all supernumerary chromosomes in the haploid nucleus prior to karyogamy, because at this stage it is not defined which of the supernumerary chromosomes will become paired and which will be unpaired. Therefore, these mechanisms should always affect all supernumerary chromosomes irrespective of whether they will become paired or unpaired after karyogamy. If such a pre-meiotic amplification or preferential segregation would occur, the diploid zygote, after karyogamy, would be trisomic for all paired supernumerary chromosomes and disomic for the unpaired chromosomes. This trisomy would result in a non-Mendelian inheritance of paired supernumerary chromosomes. We did however, observe a general Mendelian segregation pattern for the paired supernumerary chromosomes and importantly did not find any indication of additional copies of paired supernumerary chromosomes in the tetrad

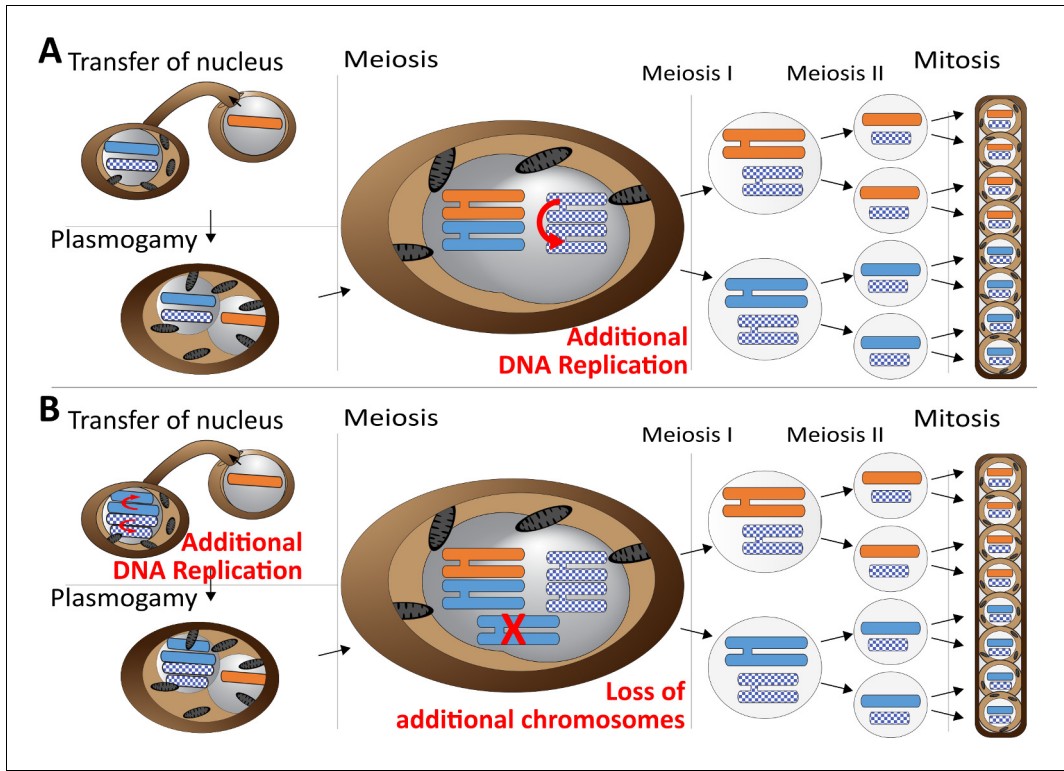

**Figure 5.** Meiotic chromosome drive in *Z.tritici*. Schematic illustration depicting two possible mechanisms for the observed meiotic chromosome drive of female derived unpaired supernumerary chromosomes. Light blue/orange: paired supernumerary chromosome. Checkered blue: unpaired supernumerary chromosome. (A) Chromosome drive occurring during meiosis. Only those unpaired supernumerary chromosomes in the zygote that originated from the female parent are subject to an additional round of DNA replication, allowing for pairing of the two copies of the chromosome. (B) Alternative scenario under which the chromosome drive occurs prior to meiosis. All supernumerary chromosomes are amplified to double the copy number during development of the female ascogonium. The supernumerary chromosomes are paired in the zygote during meiosis. Only additional copies of the supernumerary chromosomes inherited from the female are lost while the supernumerary chromosomes inherited from the male are unaffected.

DOI: https://doi.org/10.7554/eLife.40251.011

analysis. This segregation pattern would only be possible if the additional copies of the female-inherited paired supernumerary chromosomes would be lost during meiosis (*Figure 5B*).

Tightly regulated chromosome loss has been described for sex chromosomes in several insect species during embryonic development, where maternal and paternal imprinting determine the elimination of chromosomes (*Sánchez, 2014*). For *Z. tritici*, a similar mechanism would imply that all additional copies of all female-inherited supernumerary chromosomes would be eliminated during meiosis - except for the two copies of the female-inherited unpaired supernumerary chromosomes - while male-inherited supernumerary chromosomes would be unaffected. We consider this mechanism to be unlikely. In general, any pre-meiotic mechanism should affect both unpaired and paired chromosomes and therefore would require a counteracting mechanism after karyogamy that results in the general Mendelian segregation pattern observed for the paired supernumerary chromosomes. Instead, we propose that an additional amplification affecting only female-inherited supernumerary chromosomes could be a plausible scenario explaining the observed meiotic drive of supernumerary chromosomes in *Z. tritici* during sexual mating. The additional amplification of unpaired supernumerary chromosomes would require an additional initiation of DNA replication restricted to unpaired chromosomes and therefore a feedback between pairing of homologous chromosomes and DNA replication. Although DNA replication is a highly regulated process and any feedback from chromosome pairing is currently unknown, we suggest that this model is the most likely to explain the observed pattern. A form of feedback between pairing of homologous chromosomes and DNA replication has been proposed based on the effects of Spo11 - a key mediator of pairing of homologous chromosomes - on the progression of meiotic DNA replication (*Cha et al., 2000*). Although Spo11-mediated effects on DNA replication cannot explain the additional amplification of unpaired chromosomes here, it highlights a potential for a feedback between pairing of homologous sequences and DNA replication. In addition, pairing of homologous chromosomes or sequences has also been described for somatic cells in *Saccharomyces cerevisiae*, *Schizosaccharomyces pombe* and *Drosophila melanogaster* highlighting the possible existence of homolog pairing prior to DNA replication (*Burgess et al., 1999*; *Dernburg et al., 1996*; *Joyce et al., 2013*; *Scherthan et al., 1994*; *Weiner and Kleckner, 1994*). We therefore consider it possible that meiosis in *Z. tritici* involves an additional feedback mechanism that induces an additional round of amplification based on the unpaired chromosome status. We hypothesize that this would involve a tightly controlled re-initiation of DNA replication restricted to the unpaired supernumerary chromosomes inherited from the female parent in *Z. tritici*. Accordingly, we speculate that this could be reflected in the distribution and the context of origins of replication on the supernumerary chromosomes. It is furthermore interesting to speculate that the different replication patterns of the supernumerary chromosomes in *Z. tritici* could be explained by the location of these chromosomes in the nucleus. The supernumerary chromosomes appear to be associated with the nuclear envelope possibly due to their enrichment in the histone modification H3K27me3 (*Harr et al., 2015*; *Schotanus et al., 2015*). Thereby, a distinct physical location of supernumerary chromosomes could result in a different regulation of DNA replication restricted to unpaired female-derived supernumerary chromosomes.

The meiotic drive observed in *Z. tritici* differs mechanistically from previously described meiotic drives in plants, animals and fungi (*Houben, 2017*; *Jones et al., 2008*; *Lindholm et al., 2016*). To date, two general mechanisms for meiotic drive are recognized: i) gamete or spore selection and ii) preferential segregation. In i) the meiotic drive element prevents maturation or function of sperm cells or kills/disables spores that do not contain the meiotic drive element and thereby increases its relative frequency. (*Lindholm et al., 2016*). Two well-known examples are the segregation distorter (SD) haplotype in *Drosophila melanogaster*, that prevents the maturation of sperm cells not carrying the SD element (*Larracuente and Presgraves, 2012*) and the Spore killer (SK) haplotype in fungi of the genus *Neurospora* that kills those ascospores that do not contain it (*Hammond et al., 2012*). In *Z. tritici* we can exclude spore selection as a mechanistic basis for the meiotic drive because we were able to isolate complete tetrads. In ii) asymmetric cell divisions can be exploited by selfish genetic elements, which preferentially segregate into the cell that will contribute genetic material to the progeny. In monkeyflower populations the chromosomes are thought to compete according to their centromere strength during asymmetric female meiosis where only one of the four meiotic products will contribute to the progeny and three will become polar bodies (*Fishman and Saunders, 2008*; *Henikoff and Malik, 2002*; *Kursel and Malik, 2018*; *Pardo-Manuel de Villena and Sapienza, 2001*). Indeed any asymmetric cell division that generates cells that contribute to the germline can

be affected by such preferential segregation. This is exemplified by the supernumerary B chromosome of rye (*Secale cereale)* which are preferentially integrated in the generative nucleus during the asymmetric first pollen mitosis and thereby preferentially transmitted to the progeny (*Houben, 2017*). Interestingly, the centromeres of the rye B chromosomes differ from the A chromosomes highlighting the importance of centromeres for the preferential segregation of meiotic drive elements (*Houben, 2017*). In *Z. tritici*, however, the centromeres of core and supernumerary chromosomes do not appear to differ (*Schotanus et al., 2015*). The fact that the meiotic drive affects only unpaired chromosomes and that when the same chromosomes are paired, we observe no meiotic drive indicates that preferential segregation is not the underlying mechanism of meiotic drive in *Z. tritici*. Consequently, the meiotic drive mechanism observed in *Z. tritici* appears to differ from previously reported mechanisms of chromosome drive.

We observed Mendelian segregation patterns for paired supernumerary chromosomes and the unpaired supernumerary chromosomes that were inherited from the male parent. In both cases, however, there are also, in some instances, non-Mendelian patterns observable, which are the result of losses of chromosomes. Losses of supernumerary chromosomes during meiosis have been frequently described for *Z. tritici* (*Croll et al., 2013*; *Fouché et al., 2018*; *Goodwin et al., 2011*; *Wittenberg et al., 2009*) but the underlying mechanisms are unclear. Overall, the losses of supernumerary chromosomes reduce their transmission to the offspring but do not completely offset the transmission advantage of female-inherited unpaired supernumerary chromosomes amplified by the observed meiotic drive (*Figure 2—figure supplement 1C*). It is however interesting to note that supernumerary chromosomes of *Z. tritici* also are extremely frequently lost during mitosis (*Möller et al., 2018*). It is possible that the frequent losses of supernumerary chromosomes could be preventing the fixation of these chromosomes in *Z. tritici.* Therefore, chromosome losses may be an adaptive process that represent a defense mechanism against the supernumerary chromosomes.

Currently, we cannot explain why the additional amplification of the supernumerary chromosome is restricted to unpaired chromosomes inherited from the female parent. In ascomycetes, plasmogamy and karyogamy are separated by a dikaryon stage, in which the female and male nuclei are separate (*Ni et al., 2011*). Consequently, there is a temporal separation of the processes that determine uniparental inheritance of the mitochondria, nuclear inheritance, and the proposed additional amplification of the unpaired supernumerary chromosomes. Female-inherited chromosome drive in the fungus would depend on a female-derived signal that persists through plasmogamy and karyogamy when DNA amplification takes place. We currently do not know the nature of this signal, but speculate that it could be mediated by epigenetic mechanisms similar to genomic imprinting. In this context, it is interesting to note that the inheritance of the gene PDA1 located on the supernumerary chromosome 14 of *Nectria haematococca* correlates with female fertility (*Kistler and Vanetten, 1984*). It is possible that a similar meiotic drive mechanism is present in *N. haematococca.* However, in contrast, to the meiotic drive in *Z. tritici* the supernumerary chromosome 14 in *N. haematococca* appears to be preferentially inherited from the male parent. In a more recent study unpaired chr14 of *N. haematococca* showed Mendelian segregation during meiosis (*Garmaroodi and Taga, 2015*), and therefore the linkage of female fertility and supernumerary chromosome inheritance is still mechanistically unclear. The example from *N. haematococca* however underlines that inheritance of supernumerary chromosomes can be influenced by the sexual roles of the parental strains.

A meiotic chromosome drive can explain the continued maintenance of supernumerary chromosomes in *Z. tritici* despite their negative effects on fitness during host infection (*Habig et al., 2017*). In previous studies crosses between field isolates of *Z. tritici* have also shown a transmission advantage of unpaired supernumerary chromosomes (*Croll et al., 2013*; *Fouché et al., 2018*; *Wittenberg et al., 2009*). Hence, the meiotic drive of unpaired chromosomes that caused the transmission advantage observed in this study by using isogenic whole chromosome deletion strains appears to be widespread also among field isolates of *Z. tritici*. This supports the possible role of the meiotic drive in the maintenance of the supernumerary chromosomes. However, it is unclear how this type of chromosome drive can act simultaneously on several separate chromosomes. In our experiments, we found that seven of the eight supernumerary chromosomes showed drive, and we hypothesize that the drive mechanism depends on a general characteristic of the supernumerary chromosomes. Supernumerary chr14, for which no such drive was observed, is the largest of the supernumerary chromosomes (773 kb) in IPO323 (*Goodwin et al., 2011*). A large insertion spanning

approx. 400 kb in chr14 shows presence/absence polymorphism in *Z. tritici* resulting in isolates that contain a much smaller chr14 (*Croll et al., 2013*). Interestingly, the smaller chr14 showed a transmission advantage when present in one of the parental strains in a previous study (*Croll et al., 2013*), which could point to chromosome size as a factor influencing the observed drive.

In conclusion, this study shows that supernumerary chromosomes of *Z. tritici* are subjected to a meiotic drive, which is probably dependent on an additional meiotic amplification of unpaired chromosomes. This mechanism may explain the continued maintenance of supernumerary chromosomes in *Z. tritici* over long evolutionary periods in spite of their frequent loss during mitosis and their fitness cost during plant infection and asexual propagation.

## Materials and methods

### Fungal and plant material

The Dutch isolates IPO323 and IPO94269 are available from the Westerdijk Institute (Utrecht, The Netherlands) with the accession numbers CBS115943 and CBS115941. *Triticum aestivum* cultivar Obelisk used for the *in planta* fungal mating was obtained from Wiersum Plantbreeding BV (Winschoten, The Netherlands). Sexual crosses were conducted as described in (*Kema et al., 1996*; *Kema et al., 2018*).

### Fungal growth conditions

IPO94269 was maintained in liquid yeast glucose (YG) broth (30 g/L glucose and 10 g/L yeast extract) at 15°C on an orbital shaker. Due to their tendency to form hyphal lumps in liquid media IPO323 and IPO323-derived whole chromosome deletion strains and all progeny were maintained on solid YMS (4 g/L yeast extract, 4 g/L malt extract, 4 g/L sucrose, and 20 g/L agar) at 18°C. For infection, cells were washed once and diluted in $H_2O$ including 0.05% Tween20 to the indicated cell density.

### Sexual mating of *Z. tritici* strains

Sexual crosses were performed as previously described in (*Kema et al., 1996*; *Kema et al., 2018*). In short: 11–14 day old wheat plants were infected by spraying until droplets run-off the leaf surface. Plants were kept at 100% humidity for 48 hr before placing them for 12 days at 90% humidity and 16 hr light days. At day 14 post infection all except the first leaf of each plant were removed, the plants transferred to buckets with fertilized soil, and the buckets put into coarse netting, placed outside and regularly watered. Seven to eleven weeks after infection infected leaves were harvested weekly and placed in tap water over night at room temperature. The infected leaves were placed on wet filter paper occupying ¼ of a Petri dish lid, excessive water removed and a Petri dish containing 2% water agar (WA) was placed on top to collect the forcefully ejected ascospores. Every 10 min for a total of 80 min the water agar containing petri dish was rotated by 90° to collect the ejected ascospores. The WA plates were incubated for 18–24 hr at room temperature and ascospores were counted by visual inspection using a dissecting stereo microscope. A total of 39 independent crosses were conducted for this study. *Table 1* provides a summary of the crosses performed and the verified progeny obtained from these crosses.

### Ascospore isolation and ascus verification

Ascospores were isolated from the WA 18–28 hr after ejection from the ascus using a dissecting stereo microscope and a sterile syringe needle, placed onto YMS plates and grown for 6–7 days at 18°C. One colony per isolated ascospore was streaked out on YMS plates and grown again for 6–7 days at 18°C. Single colonies were isolated and used for all further characterization. To isolate all ascospores from an ascus eight germinated ascospores that were spatially separate from other ascospores on the WA were considered to be ejected from one ascus. All eight ascospores were isolated.

To verify that the eight ascospores originated from the same ascus, the following six segregating markers located on the essential chromosomes were used: *mat1-1/mat1-2* (*Waalwijk et al., 2002*), 11O21, 04L20, caa-0002, ggc-001 and ac-0001 (*Goodwin et al., 2007*) (*Table 2*).

**Table 1.** Summary of crosses and progeny generated in this study.

| # | Parental strain 1 | Parental strain 2 | Unpaired chr from IPO94269 | Unpaired chr from IPO323 | Condition | Ascospores (random[*]/all) | | | Verified tetrads (mtIPO323/ mtIPO94269) | | |
|---|---|---|---|---|---|---|---|---|---|---|---|
| | | | | | | Exp A | Exp B | Exp C | Exp A | Exp B | Exp C |
| 1 | IPO323 | IPO94269 | - | chr18, chr20 | Co-inoculation $10^7$ cells/mL | 96/96 | 12/96 | 51/88 | - | 2/2 | 0 |
| 2 | IPO323 Δchr14 | IPO94269 | chr14 | chr18, chr20 | Co-inoculation $10^7$ cells/mL | 96/96 | 8/64 | 38/72 | - | 3/0 | 1/0 |
| 3 | IPO323 Δchr21 | IPO94269 | chr21 | chr18, chr20 | Co-inoculation $10^7$ cells/mL | 89/89 | 4/32 | 52/78 | - | 0 | 0 |
| 4 | IPO323 Δchr16 | IPO94269 | chr16 | chr18, chr20 | Co-inoculation $10^7$ cells/mL | 96/96 | 6/48 | 38/115 | - | 1/0 | 0/1 |
| 5 | IPO323 Δchr17 | IPO94269 | chr17 | chr18, chr20 | Co-inoculation $10^7$ cells/mL | 96/96 | 2/16 | 15/59 | - | 0 | 2/0 |
| 5 | IPO323 Δchr19 | IPO94269 | chr19 | chr18, chr20 | Co-inoculation $10^7$ cells/mL | 96/96 | 9/72 | 38/77 | - | 3/1 | 0/1 |
| 7 | IPO323 Δchr20 | IPO94269 | - | chr18 | Co-inoculation $10^7$ cells/mL | 96/96 | 4/32 | 19/31 | - | 2/0 | 0 |
| 8 | IPO323 Δchr18 | IPO94269 | - | chr20 | Co-inoculation $10^7$ cells/mL | 96/96 | 4/32 | 30/67 | - | 1/0 | 2/0 |
| 9 | IPO323 Δchr15 | IPO94269 | chr15 | chr18, chr20 | Co-inoculation $10^7$ cells/mL | 96/96 | 6/48 | 14/54 | - | 1/0 | 0/1 |
| 10 | IPO323 Δchr19 | IPO94269 | chr19 | chr18, chr20 | IPO323 $10^6$ cells/mL | - | - | 25/42 | - | - | 0 |
| 11 | IPO323 Δchr19 | IPO94269 | chr19 | chr18, chr20 | IPO323 $10^5$ cells/mL | - | - | 38/43 | - | - | 0 |
| 12 | IPO323 Δchr19 | IPO94269 | chr19 | chr18, chr20 | IPO323 $10^4$ cells/mL | - | - | 2/16 | - | - | 0 |
| 13 | IPO323 Δchr19 | IPO94269 | chr19 | chr18, chr20 | IPO94269 +6dpi | - | - | 40/46 | - | - | 0 |
| 14 | IPO323 Δchr19 | IPO94269 | chr19 | chr18, chr20 | IPO94269 +12dpi | - | - | 11/11 | - | - | 0 |
| 15 | IPO323 Δchr19 | IPO94269 | chr19 | chr18, chr20 | IPO94269 $10^6$ cells/mL | - | - | 60/84 | - | - | 0 |
| 16 | IPO323 Δchr19 | IPO94269 | chr19 | chr18, chr20 | IPO94269 $10^5$ cells/mL | - | - | 48/80 | - | - | 0 |
| 17 | IPO323 Δchr19 | IPO94269 | chr19 | chr18, chr20 | IPO94269 $10^4$ cells/mL | - | - | 28/28 | - | - | 0 |
| 18 | IPO323 Δchr19 | IPO94269 | chr19 | chr18, chr20 | IPO323 +6pdi | - | - | 5/29 | - | - | 0 |
| 19 | IPO323 Δchr19 | IPO94269 | chr19 | chr18, chr20 | IPO323 +12dpi | - | - | 51/71 | - | - | 1/0 |
| | | | | | $\Sigma$ | 761/761 | 55/440 | 603/1091 | - | 13/3 | 5/3 |

[*]Includes random and randomized ascospores. Randomized ascospores were generated by randomly selecting one ascospore per tetrad.

DOI: https://doi.org/10.7554/eLife.40251.012

**Table 2.** Overview of six segregating markers located on the essential chromosomes.

| Marker | Localization (IPO323) | Localization (IPO94269) | Primer1 | Primer 2 | Product size in IPO323 [bp] | Product size in IPO94269 [bp] |
|---|---|---|---|---|---|---|
| mat1-1/mat1-2 | chr13 621930–622924 | Unitig11 627136–627792 | MAT1-1F, MAT1-1R | MAT1-2F, MAT1-2R | 340 | 660 |
| 11O21 | chr4 642791–642996 | Unitig03 757794–757990 | 11O21F | 11O21R | 205 | 199 |
| 04L20 | chr4 2276298–2276497 | Unitig03 2427163–2427370 | 04L20F | 04L20R | 192 | 199 |
| caa-0002 | chr3 2927294–2927704 | Unitig02 566646–567041 | 2996 | 2997 | 412 | 396 |
| ggc-001 | chr5 1190388–1190640 | Unitig04 1143568–1143802 | 2998 | 2999 | 254 | 234 |
| ac-0001 | chr7 266661–266847 | Unitiq05 372979–373151 | 3000 | 3001 | 187 | 173 |

DOI: https://doi.org/10.7554/eLife.40251.013

In *Z. tritici* as well as in other ascomycetes meiosis I and meiosis II is followed by an additional mitotic cell division (*Wittenberg et al., 2009*). This mitotic division will result in eight ascospores which consist of four pairs of identical twins. Therefore we considered only those tetrads to be correctly isolated that showed a 4/4 ratio of each of the six core chromosomal markers AND each ascospore having exactly one twin. Tetrads where an ascospore had none or more than one twin were excluded even if they showed a 4/4 ratio of all core chromosomal markers described in *Table 2*. As an internal PCR control an amplification of the *gapdh* (primer: 879, 880) was used to verify negative PCR results.

In experiment B a total of 440 ascospores were isolated from 55 asci, and of these, 128 ascospores from 16 asci met the criteria and were considered to be derived from within one ascus. In experiment C a total of 504 ascospores were isolated from a total of 63 asci, and of these, 64 of eight asci met these criteria. The mitochondrial genotype was determined using the primers Mt-SSR-F and Mt-SSR-R (*Kema et al., 2018*).

Ascospores were karyotyped for the presence of supernumerary chromosomes using primer pairs specifically designed to test for the presence of chromosomes 14, 15, 16, 17, 18, 19, 20, and 21 derived from IPO323 and from IPO94269 (*Supplementary file 1*, *Figure 3—source data 1*) using standard conditions (*Habig et al., 2017*). For unpaired chromosomes that were overrepresented in the ascospores of experiment B and C, we validated their presence by two additional PCRs that amplify a sequence in the right subtelomeric and left subtelomeric regions of the chromosome (*Supplementary file 1*, *Figure 3—source data 1*). For paired supernumerary chromosomes, we designed the primers to reveal a size difference in the amplification products of IPO323-derived and IPO94269-derived chromosomes (*Supplementary file 1*, *Figure 3—source data 1*).

## Statistical analysis

All statistical analyses were conducted in R (version R3.4.1) (*R Core Team, 2015*) using the suite R Studio (version 1.0.143) (*Team RStudio, 2015*). Two-sided Fisher's exact tests were performed at a confidence level of 0.95. Due to the codependency of mitochondrial data of ascospores isolated from a potential or verified ascus, because all ascospores of an ascus will receive the same mitochondrial genotype, one ascospore from each potential and verified ascus was randomly selected and included in the statistical analysis of the mitochondrial genotype transmission. For large datasets the Fisher's exact test was replaced by the Pearson's Chi-squared test. Two-sided binomial tests were performed with a hypothesized probability of $p=0.5$ at a confidence level of 0.95 in all statistical analyses to determine a deviation of an assumed Mendelian segregation of unpaired supernumerary chromosomes which would be predicted to be present in 50% of the progeny. All statistical analyses on transmission of supernumerary chromosomes included all randomly selected ascospores as well as all ascospores selected from potential and verified asci.

## Genome sequencing

For sequencing, DNA of IPO94269 and 16 ascospores was isolated using a phenol-chloroform extraction protocol as described previously (*Sambrook and Russell, 2001*). Library preparation and sequencing using a Pacific Biosciences Sequel for IPO94269 and Illumina HiSeq3000 machine for 16 ascospore-derived colonies were performed at the Max Planck-Genome-centre, Cologne, Germany. Reads have been deposited in the Sequence Read Archive and are available under the BioProject PRJNA438050. Assembly of the IPO94269 genome was conducted at the Max Planck-Genome-center, Cologne using the software suite HGAP 4 (*Chin et al., 2016*) from Pacific Biosciences with the default settings. Synteny analysis was conducted with SyMAP version 4.2 (*Soderlund et al., 2006*; *Soderlund et al., 2011*) (*Figure 1—figure supplement 1*). Illumina reads of the ascospores were filtered and mapped to the reference genome of IPO323 (*Goodwin et al., 2011*) as previously described (*Habig et al., 2017*) in which the transposable elements were masked (*Grandaubert et al., 2015*).

## Reference mapping and SNP calling of Illumina reads

Paired-end reads of 150 bp were mapped directly to the genome of the reference isolate IPO323 (*Goodwin et al., 2011*). Processing of the reads was carried out using the below listed pipeline:

1. Quality filtering using Trimmomatic V0.30 (*Bolger et al., 2014*)
   java -jar/trimmomatic−0.30.jar PE -phred33 R1.fastq R2.fastq R1_paired.fastq R1_unpaired.fastq R2_paired.fastq R2_unpaired.fastq HEADCROP:2 CROP:149 LEADING:3 TRAILING:3 SLIDINGWINDOW:4:15 MINLEN:50 http://www.usadellab.org/cms/?page=trimmomatic
2. Mapping to IPO323 reference genome using Bowtie 2 version 2.1.0 (*Langmead and Salzberg, 2012*)
   bowtie2 -p 6 -q -x IPO323_reference −1 R1_paired.fastq −2 R2_paired.fastq -S R.sam http://bowtie-bio.sourceforge.net/bowtie2/index.shtml
3. Converting to BAM and sorting using Picard 1.141
   java -jar/picard.jar SortSam INPUT = R.sam OUTPUT = R.bam SORT_ORDER = coordinate http://broadinstitute.github.io/picard
4. SNP calling using samtools (version1.7) and bcftools (version 1.6)
   samtools mpileup -E -C50 -Q20 -q20 -uf IPO323_reference R.bam | bcftools call –ploidy-file ploidy.txt -vc -O u -o R.raw.bcf

## IPO94269 specific SNP calling using PacBio reads

Pacific Biosciences Sequel reads for IPO94269 were mapped onto the IPO323 reference genome (*Goodwin et al., 2011*) using the below listed pipeline:

1. Mapping to IPO323 reference genome using BLASR (version 5.3.2)
   blasr A.subreads.bam IPO323_reference.fasta –bam –nproc 16 –out A _aligned.bam
2. Calling variants using arrow (version 2.3.2)
   arrow -j10 A_aligned.sorted.bam -r IPO323_refernce.fasta -o A_variants.vcf -o A_consensus.fasta
3. Detection of shared SNP between ascospores and parent IPO94269 using bcftools (version 1.6)
   bcftools isec -p/output_dir/A_variants.bcf.gz B_variants.bcf.gz

## Acknowledgments

The authors thank Michael Freitag and members of the Environmental Genomics group for helpful discussions and Diethard Tautz for comments to a previous version of this manuscript. The study was funded by a personal grant to EHS from the State of Schleswig Holstein and the Max Planck Society. The funders had no role in study design, data collection and interpretation, or the decision to submit the work for publication.

## Additional information

### Funding

| Funder | Grant reference number | Author |
|---|---|---|
| State of Schleswig Holstein | | Eva Holtgrewe Stukenbrock |
| Max-Planck-Gesellschaft | Open-access funding | Eva Holtgrewe Stukenbrock |

The funders had no role in study design, data collection and interpretation, or the decision to submit the work for publication.

### Author contributions

Michael Habig, Conceptualization, Formal analysis, Investigation, Visualization, Writing—original draft; Gert HJ Kema, Conceptualization, Supervision, Writing—review and editing; Eva Holtgrewe Stukenbrock, Conceptualization, Supervision, Funding acquisition, Project administration, Writing—review and editing

### Author ORCIDs

Michael Habig http://orcid.org/0000-0002-8059-806X
Gert HJ Kema http://orcid.org/0000-0002-2732-6911
Eva Holtgrewe Stukenbrock http://orcid.org/0000-0001-8590-3345

### Decision letter and Author response

Decision letter https://doi.org/10.7554/eLife.40251.025
Author response https://doi.org/10.7554/eLife.40251.026

## Additional files

### Supplementary files

• Supplementary file 1. All primers used in this study.
DOI: https://doi.org/10.7554/eLife.40251.014

• Supplementary file 2. Summary of all PCR marker results for experiment A.
DOI: https://doi.org/10.7554/eLife.40251.015

• Supplementary file 3. Summary of all PCR marker results for experiment B.
DOI: https://doi.org/10.7554/eLife.40251.016

• Supplementary file 4. Summary of all PCR marker results for experiment C.
DOI: https://doi.org/10.7554/eLife.40251.017

• Supplementary file 5. Summary of statistical tests performed in this study.
DOI: https://doi.org/10.7554/eLife.40251.018

• Supplementary file 6. Frequency of transmission of paired chromosomes to progeny ascospores.
DOI: https://doi.org/10.7554/eLife.40251.019

• Transparent reporting form
DOI: https://doi.org/10.7554/eLife.40251.020

### Data availability

Sequencing reads have been deposited in the Sequence Read Archive and are available under the BioProject PRJNA438050. All data generated or analysed during this study are included in the manuscript and supporting files (Supplementary files 2-4). Source data files have been provided for Figure 2 and Figure 2—figure supplement 1

The following dataset was generated:

| Author(s) | Year | Dataset title | Dataset URL | Database and Identifier |
|---|---|---|---|---|
| Michael Habig | 2018 | Tetrad-Anaylsis of Zymoseptoria tritici Ascospore | https://www.ncbi.nlm. nih.gov/bioproject/ | NCBI BioProject, PRJNA438050 |

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
