## [Decision Letter]

Thank you for submitting your article "Meiotic drive of female-inherited supernumerary chromosomes in a pathogenic fungus" for consideration by *eLife*. Your article has been reviewed by three peer reviewers, one of whom is a member of our Board of Reviewing Editors, and the evaluation has been overseen by Detlef Weigel as the Senior Editor. The reviewers have opted to remain anonymous.

The reviewers have discussed the reviews with one another and the Reviewing Editor has drafted this decision to help you prepare a revised submission.

Summary:

Several fungal species contain supernumerary (or accessory) chromosomes – these are not required for growth in standard laboratory conditions and can sometimes segregate in the absence of a paired homolog. The authors used isogenic strains of *Zymoseptoria tritici* to assess the inheritance of unpaired supernumerary chromosomes. By performing a series of crosses, the authors found that the segregation of unpaired supernumerary chromosomes was preferentially inherited from the female parent, identified by the mitochondrial genotype. It was found that the cell density of the parent had a significant influence on which parent served as the female, where the less dense served as the female. Those unpaired chromosomes inherited from the female showed meiotic drive. Paired supernumerary chromosomes showed Mendelian ratios with occasional chromosomal losses/rearrangements. The meiotic drive was postulated to be the result of DNA replication of unpaired chromosomes from the female parent that could be dependent on a characteristic of the supernumerary chromosomes, which would represent a new type of meiotic drive. Although the study lacks the identification of a mechanism, the discovery is original and novel and will be of great interest to the field. We invite you to add a bit to your speculation at the end about the nature of the signal and the potential epigenetic processes that might be involved.

Essential revisions:

1) A linkage between the preferential inheritance of supernumerary chromosomes and female fertility has been described before in *Nectria haematococca* (Kistler and VanEtten, 1984), although at the time it was not known that the gene responsible for the PDA phenotype resided on a supernumerary chromosome. Please include this reference and discuss it in the context of your findings.

2) The experiments were well conducted and there was a significant number of ascospores analyzed to base the conclusions. However, the authors used isogenic strains. Can the authors comment on how the use of isogenic lines may influence interpretation of how common this phenomenon may be for *Z. tritici* supernumerary chromosomes in the field?

3) There was confusion about how the genotyping was done. For the tetrads, six 'segregating' markers were used to determine if the groups of 8 spores really resulted from meiosis. Are those markers all unlinked from each other? Also, the expected sizes of the PCR products only differed by a few base pairs. Is that how the PCR products were distinguished, or were they sequenced? If they were distinguished by size, please show an example gel. The Materials and methods also say approving a tetrad required 'exactly two ascospores were twins.' Did you mean that each ascospore had to have a twin?

4) The PFGE karyotyping was not convincing without Southern Blotting to demonstrate the presence of a given chromosome in progeny. The chromosomes were not resolved well on the gel and recombination between chromosomes appears to affect the size of the chromosomes in the progeny.

5) In Figure 2, please show the transmission data for each chromosome rather than the pooling of the data from different chromosomes.

6) For Figure 2, please use box and whiskers plots (or any other type of plot that showed not just the average value but also the variance of the individual measurements around the average) to illustrate your results.

7) Were the parental strains inoculated at the same time and the same densities for experiments A, B, and C? If those used the same cell densities and timing, there is a lot of variation not explained by those things.

8) The conclusion that the later inoculated strain takes on the female role was not well supported by the data. In Supplementary Figure 2D, inoculating the IPO323 parent 6hrs later had a minimal effect on the probability that it transmitted mitochondria. In addition, inoculating IPO323 12hrs later resulted in even less transmission of mitochondria. Furthermore, based on the last paragraph of the subsection “Transmission of mitochondria is affected by the cell density”, it appears that your experiment on infection timing is hard to interpret given that you couldn't adequately control for cell density. Given that the design of this experiment seems to have been suboptimal (at least a posteriori), one suggestion would be to remove it from the manuscript and just present the data on cell density.

9) How did SNPs between the parental strains segregate in the sequenced tetrads? It would be good to see if they show Mendelian transmission on the core chromosomes and paired supernumerary chromosomes.

10) The authors do not make much about the drag of unpaired chromosomes when inherited from the male parent. This too is non-Mendelian transmission and is also important. This result also warrants highlighting/discussion.

11) More discussion comparing this system to previously described types of drivers to highlight that this represents a novel type of meiotic drive is warranted. Along similar lines, please speculate a bit more what the underlying mechanisms could be and make it clear that/how you think about such issues.

12) Subsection “Transmission of mitochondria is affected by the cell density”, first paragraph: Please provide a brief justification why cell density and relative timing were considered.

13) Discussion, fourth paragraph: How many components of fitness were measured in the Habig et al. (2017) paper? It could very well be the case that supernumerary chromosomes negatively impact fitness in certain conditions but positively impact fitness in others.

14) There were quite a few typos and minor errors in the manuscript – please proofread thoroughly before submitting the revised version.

[Editors' note: further revisions were requested prior to acceptance, as described below.]

Thank you for submitting your article "Meiotic drive of female-inherited supernumerary chromosomes in a pathogenic fungus" for consideration by *eLife*. Your article has been reviewed by three peer reviewers, one of whom is a member of our Board of Reviewing Editors, and the evaluation has been overseen by Detlef Weigel as the Senior Editor. The reviewers have opted to remain anonymous.

The reviewers have discussed the reviews with one another and the Reviewing Editor has drafted this decision to help you prepare a revised submission.

Summary:

In this revised manuscript, the authors have adequately addressed the concerns that were raised from the previous submission. The specific break down of individual supernumerary chromosomes (Figure 2) demonstrating meiotic drive for all the supernumerary chromosomes (except 14, which was further elaborated on in the Discussion) and the explanation of the influence of environmental factors in the sexual roles of the crosses were welcome additions. The Discussion section was significantly revised and more informative than the previous submission. Importantly, the additions made to the manuscript improve the significance of the research. In particular, the addition the speculation concerning the details of the meiotic drive and the paragraph of concerning the comparison between this meiotic drive and those previously described. Overall, the manuscript is improved and the experimental approach well conducted.

Essential revisions:

The authors add new analysis of their octad sequencing data. They say that the 'vast majority' of SNPs show Mendelian segregation for the core and paired supernumerary chromosomes, but then report that ~32% of SNPs did not show this pattern. I do not consider 70% the 'vast majority.' I also have a hard time ignoring that 32% non-Mendelian transmission, given the topic of this paper. I need to know the distribution of those SNPs. Are they clustered, or are they distributed amongst other Mendelian SNPs? Also, what was the read coverage for those SNPs? Can we safely ignore them based on poor coverage? They argue that failure to detect the SNPs is most likely because they expected absent SNPs to be present in pairs based on post-meiotic mitosis. However, post-meiotic segregation can occur when heteroduplex DNA is not repaired in meiosis. Given the other non-standard stuff observed in this species, I think this sequence data must be thoroughly interrogated and explained.

---

## [Author Response]

Essential revisions:1) A linkage between the preferential inheritance of supernumerary chromosomes and female fertility has been described before in Nectria haematococca (Kistler and VanEtten, 1984), although at the time it was not known that the gene responsible for the PDA phenotype resided on a supernumerary chromosome. Please include this reference and discuss it in the context of your findings.

We thank the reviewer for their comment. This information has escaped our attention. We have included the following arguments in our Discussion:

“In this context, it is interesting to note that the inheritance of the gene PDA1 located on the supernumerary chromosome 14 of *Nectria haematococca* correlates with female fertility (Kistler and VanEtten, 1984). […] The example from *N. haematococca* however underlines that inheritance of supernumerary chromosomes can be influenced by the sexual roles or the parental strains.”

2) The experiments were well conducted and there was a significant number of ascospores analyzed to base the conclusions. However, the authors used isogenic strains. Can the authors comment on how the use of isogenic lines may influence interpretation of how common this phenomenon may be for Z. tritici supernumerary chromosomes in the field?

The meiotic drive described in our experiments results in a transmission advantage of unpaired accessory chromosomes. A similar transmission advantage of unpaired supernumerary chromosomes has been described for crosses between field isolates of *Z. tritici* (Croll et al., 2013; Fouché et al., 2018; Wittenberg et al., 2009), which indicates that the meiotic drive is widespread among field isolates. We have extended the description of the previous experiments to better reflect that the observed transmission advantage is also often observed during crosses of field isolates. We have also included following statement in the Discussion:

“In previous studies crosses between field isolates of *Z. tritici* have also shown a transmission advantage of unpaired supernumerary chromosomes (Croll et al., 2013; Fouché et al., 2018; Wittenberg et al., 2009). […] This supports the possible role of the meiotic drive in the maintenance of the supernumerary chromosomes.”

3) There was confusion about how the genotyping was done. For the tetrads, six 'segregating' markers were used to determine if the groups of 8 spores really resulted from meiosis. Are those markers all unlinked from each other?

We are sorry for the unclear explanation. The six segregating markers that are used for the genotyping are located on five different core chromosomes which can be assumed to segregate independently from each other. Two markers located on one chromosome (11O21 and 04L20) but are separated by 1.6 Mb. *Z. tritici* has an exceptionally high recombination rate of 66 cM/Mb genomic average (Stukenbrock and Dutheil, 2017). No significant linkage between the marker 11O21 and 04L20 were detectable in our experiments (Fisher’s exact test, p=0.11). Therefore we concluded that the six marker are unlinked from each other. We have added the information of the location of the markers in Table 2. In addition we have included the additional Figure 3—figure supplement 2 which depicts the occurrence of IPO94269 specific SNPs in the eight ascospores of an ascus for both tetrads that were sequenced completely. The observed pattern allows the detection of crossover. All eight ascospores for each ascus were included in reciprocal crossover and therefore can only be the result of a single meiotic event. This verifies the suitability of using six segregating markers on core chromosomes to identify complete tetrads.

Also, the expected sizes of the PCR products only differed by a few base pairs. Is that how the PCR products were distinguished, or were they sequenced? If they were distinguished by size, please show an example gel.

The PCR products were distinguished by gel electrophoresis. We have included a representative gel electrophoresis picture for each of these PCR products in Figure 3—figure supplement 3. The analysis of these PCR-based markers by gel electrophoresis in IPO323 and IPO94269 was conducted as previously described (Kema et al., 2018; Goodwin et al., 2007).

The Materials and methods also say approving a tetrad required 'exactly two ascospores were twins.' Did you mean that each ascospore had to have a twin?

To clarify this point, we have included the following statement in the Materials and methods section:

“In *Z. tritici* as well as in other ascomycetes meiosis I and meiosis II is followed by an additional mitotic cell division (Wittenberg et al., 2009). […] Tetrads where an ascospore had none or more than one twin were excluded even if they showed a 4/4 ratio of all core chromosomal markers described in Table 2.”

4) The PFGE karyotyping was not convincing without Southern Blotting to demonstrate the presence of a given chromosome in progeny. The chromosomes were not resolved well on the gel and recombination between chromosomes appears to affect the size of the chromosomes in the progeny.

In those instances where meiotic drive was detected all PCR-based markers as well as the results of whole genome sequencing showed the presence of female-inherited unpaired chromosomes in all eight ascospores of a tetrad although only one parental strain contained the chromosomes. The analysis of the inheritance pattern of IPO94269 specific SNPs to the progeny

(Figure 3—figure supplement 2) also highlights that the core chromosomes and paired supernumerary chromosomes show a Mendelian transmission pattern whereas genetic material of unpaired female-inherited chromosomes are present in all eight ascospores of a tetrad. We therefore consider all the conclusion presented in the manuscript to be supported by the presented data. In our opinion a Southern Blot is not required for our conclusion.

5) In Figure 2, please show the transmission data for each chromosome rather than the pooling of the data from different chromosomes.

In Figure 2B we have included the transmission data for each unpaired chromosomes. This allows us to highlight that the meiotic drive affects seven of the eight unpaired chromosomes with only chromosome 14 showing a different transmission pattern.

6) For Figure 2, please use box and whiskers plots (or any other type of plot that showed not just the average value but also the variance of the individual measurements around the average) to illustrate your results.

All panels in Figure 2 summarize count data for two categories (IPO94269 mtDNA/IPO323 mtDNA for panel A/C and chr. present/chr. absent for panel B and D). There is no meaningful average/median and variance available for count data which is why we choose to display relative frequencies of the count data.

7) Were the parental strains inoculated at the same time and the same densities for experiments A, B, and C? If those used the same cell densities and timing, there is a lot of variation not explained by those things.

Yes, the parental strains were inoculated at the same time relative to the host plant germination and the same cell densities were used in all three experiments. However, successful crosses were dependent on an incubation step outside of the greenhouses/phytochamber as in the originally described protocol for sexual crosses of *Z. tritici* (Kema et al., 1996). Therefore, the environmental conditions could not be controlled during the three experiments. The three independent experiments were conducted at different time points during the course of the year (Exp.A: December-March, Exp.B: April- July, Exp.C: September –December). The outcome of these three experiments varied mainly in the relative frequencies at which the two parental strains were female (Figure 2A). In this study, we show that cell density affects the sexual role of the two parental strains (Figure 2C). We therefore hypothesize that the high variability between the three experiments in the sexual role of the two parental strains is mainly caused by two parental strains being differentially affected by the different environmental conditions during the different seasons. In contrast, extremely low variation was observed between three experiments once data was analyzed separately according to which of the two parental strains was female. We have included the following statement in the manuscript to hypothesize on the origin of the variation in sexual roles between the three experiments:

“We hypothesize that variability in environmental factors between the three experiments, which were conducted during different seasons and at different locations, may explain the observed variability in the frequency of sexual roles of the two strains among the three experiments. Accordingly, the transmission of unpaired supernumerary chromosomes may also be affected by seasonal changes in environmental factors.”

8) The conclusion that the later inoculated strain takes on the female role was not well supported by the data. In Supplementary Figure 2D, inoculating the IPO323 parent 6hrs later had a minimal effect on the probability that it transmitted mitochondria. In addition, inoculating IPO323 12hrs later resulted in even less transmission of mitochondria. Furthermore, based on the last paragraph of the subsection “Transmission of mitochondria is affected by the cell density”, it appears that your experiment on infection timing is hard to interpret given that you couldn't adequately control for cell density. Given that the design of this experiment seems to have been suboptimal (at least a posteriori), one suggestion would be to remove it from the manuscript and just present the data on cell density.

We have removed the data on the effect of the timing of inoculation on the sexual role of the two parental strains.

9) How did SNPs between the parental strains segregate in the sequenced tetrads? It would be good to see if they show Mendelian transmission on the core chromosomes and paired supernumerary chromosomes.

We have analyzed the transmission of IPO94269 specific SNPs to the progeny and included these new results that are illustrated in Figure 3—figure supplement 2A-F. Here, we map the transmission of parental IPO94269 specific SNPs to the progeny in two complete tetrads (eight ascospores each for ascus A03-4 and A08-1). On each ascospore IPO94269 haplotypes alternate with IPO323 haplotypes that are devoid of IPO94269 SNPs. The distribution of these haplotypes among the eight ascospores allows for the identification of recombination events that resulted in crossover. These crossover events are detectable within all eight ascospores indicating that different sister chromatids were involved in the individual recombination events. At any given location within the genome a maximum of four ascospores contain the IPO94269 haplotype indicating Mendelian segregation. Additionally we have detected the transmission of individual IPO94269 specific SNPs. For both tetrads the majority of the SNPs located on the core and paired supernumerary chromosomes are present in four of the eight progeny ascospores indicating Mendelian segregation of these SNPs. This pattern is very similar between core and paired supernumerary chromosomes. A number of SNPs occur in three, two or one ascospores. We hypothesize that this is due to low sensitivity in detecting SNPs. Very few SNPs are present in more than four ascospores (69 of a total of 167346 SNPs for ascus A03-4, 130 of a total of 170537 SNPs for ascus A08-1) and we speculate that these are the result of gene conversion. Based on these analyses we conclude that the vast majority of SNPs on core and paired accessory chromosomes show Mendelian segregation and have included the following statement in the Results section:

“Whole genome sequencing of two tetrads allowed for the dissection of the inheritance of SNPs on paired supernumerary and core chromosomes. […] In conclusion, a similar pattern of the transmission of SNPs was detected for core chromosomes and paired supernumerary chromosomes consistent with Mendelian segregation for the vast majority of SNPs.”

10) The authors do not make much about the drag of unpaired chromosomes when inherited from the male parent. This too is non-Mendelian transmission and is also important. This result also warrants highlighting/discussion.

We have addressed the drag of the unpaired supernumerary chromosomes inherited from the male in a new separate paragraph in the Discussion:

“We observed Mendelian segregation patterns for paired supernumerary chromosomes and the unpaired supernumerary chromosomes that were inherited from the male parent. […] Therefore, chromosome losses may be an adaptive process that represent a defense mechanism against the supernumerary chromosomes.”

11) More discussion comparing this system to previously described types of drivers to highlight that this represents a novel type of meiotic drive is warranted. Along similar lines, please speculate a bit more what the underlying mechanisms could be and make it clear that/how you think about such issues.

We have included a new paragraph comparing the meiotic drive of *Z. tritici* with previously described mechanisms of meiotic drives in the Discussion. We have also included a brief speculation on the underlying mechanism for the regulated re-initiation of DNA replication but have limited ourselves, because at this stage we feel that this is highly speculative.

Due to the extent of the changes we kindly refer directly to the Discussion section of the manuscript.

12) Subsection “Transmission of mitochondria is affected by the cell density”, first paragraph: Please provide a brief justification why cell density and relative timing were considered.

Our justification was our previous observation showing a density-dependent effect on the timing of the switch to necrotic growth (M. Habig, unpublished results) and a recent finding on the effect of cell density on the competition between sexual and asexual modes of reproduction for *Z. tritici*. Both findings highlight the effect of cell density on biological processes. We have therefore included the following sentences in the manuscript:

“Recently, competition between sexual and asexual modes of reproduction in *Z. tritici* was shown to be affected by the cell density (Suffert et al., 2018). We therefore hypothesized that the cell density of the two parental strains could also affect the sexual roles of the two parental strains.”

13) Discussion, fourth paragraph: How many components of fitness were measured in the Habig et al. (2017) paper? It could very well be the case that supernumerary chromosomes negatively impact fitness in certain conditions but positively impact fitness in others.

The fitness component measured in Habig et al. (2017) was the ability to produce pycnidia, the asexual fruiting bodies giving rise to pycnidiospores, in planta. Supernumerary chromosome 14, 16, 18, 19 and 21 had a significant negative effect on the ability to produce pycnidia in one wheat cultivar but did not show a significant effect on three other wheat cultivars. Therefore, Habiget al. (2017) concludes that the supernumerary chromosomes do have a host-genotype dependent fitness costs in planta. However, the experiments were conducted in controlled environments. It is correct that a positive effect of the supernumerary chromosomes under different conditions cannot be excluded. We would argue that it is impossible to test for a fitness effect in all conceivable conditions. However, the fitness costs described in *Z. tritici* should lead to an elimination of the supernumerary chromosomes if no counteracting selection applies. We argue that this could be either a function of the supernumerary chromosomes during other conditions (not tested in this manuscript) or a transmission advantage due to meiotic drive as described here.

14) There were quite a few typos and minor errors in the manuscript – please proofread thoroughly before submitting the revised version.

We have proofread the manuscript and corrected all typos/errors.

[Editors' note: further revisions were requested prior to acceptance, as described below.]

Essential revisions:The authors add new analysis of their octad sequencing data. They say that the 'vast majority' of SNPs show Mendelian segregation for the core and paired supernumerary chromosomes, but then report that ~32% of SNPs did not show this pattern. I do not consider 70% the 'vast majority.' I also have a hard time ignoring that 32% non-Mendelian transmission, given the topic of this paper. I need to know the distribution of those SNPs. Are they clustered, or are they distributed amongst other Mendelian SNPs? Also, what was the read coverage for those SNPs? Can we safely ignore them based on poor coverage? They argue that failure to detect the SNPs is most likely because they expected absent SNPs to be present in pairs based on post-meiotic mitosis. However, post-meiotic segregation can occur when heteroduplex DNA is not repaired in meiosis. Given the other non-standard stuff observed in this species, I think this sequence data must be thoroughly interrogated and explained.

We thank the reviewer for their comment and the suggestion to focus on the sequencing read coverage of the SNPs.

We analyzed the distribution of the SNPs along the genome (see Author response image 1 depicting the results for all eight ascospores of ascus A03-4 as an example). The 32% of SNPs that had less than four occurrences among the eight ascospores showed a very similar distribution within the genome as the SNPs that were found at exactly in four occurrences. There appears to be no clustering of SNPs with less than four occurrences as would be expected if a mechanistic or genetic basis (e.g. a non-Mendelian transmission of different parts of paired chromosomes) was responsible for the non-Mendelian pattern. The SNPs with less than four occurrences are interspersed between SNPs with exactly four occurrences. Therefore, the SNPs that show a non-Mendelian pattern (i.e. having less than four occurrences) are located in the vicinity of or are interspersed between SNPs that showed Mendelian segregation (i.e. having exactly four occurrences among the eight ascospores).

**Author response image 1. respfig1:** IPO94269 specific SNPs with less than four occurrences among the eight ascospores are interspersed among IPO94269 specific SNPs with exactly four occurrences among the eight ascospores. A) Whole genome view of all eight ascopsores of ascus A03-4. Stretches of IPO94269 haplotype alternate with stretches of IPO323 haplotype. Crossover events are discernible between all eight ascospores. The distribution of IPO94269 specific SNPs with less than four occurrences among the eight ascospores (blue) is similar to the distribution of the SNPs with exactly four occurrences among the eight ascospores (red). There is no separate clustering of the SNPs showing non-Mendelian inheritance. B) Detailed view of the distribution of SNPs within a representative 200 kb of core chromosome 1 and paired supernumerary chromosome 14 of all eight ascospores of ascus A03-4. IPO94269 specific SNPs with less than four occurrences among the eight ascospores are interspersed among those IPO94269 specific SNPs that show present in four of the eight ascospores.

In order to determine whether the high proportion of SNPs (32%) that were detected in less than four ascospores were due to a data quality issue, we modified our criteria for SNP detection. We argue that the probability of incorrect SNPs should decrease in regions with high coverage, and therefore redid our analysis only for regions with a higher coverage. When restricting the analysis to those SNPs that were detected in regions with > 8X coverage in at least one ascospore, the proportion of SNPs detected in exactly four ascospores increase considerably to 94% in both fully sequenced octads. We further increased the fidelity of the SNP detection by restricting our analysis to SNPs that were detected at a sequencing read coverage of > 8X in all ascospores. Thereby we find a considerably higher proportion of SNPs showing Mendelian segregation, 97% and 98% (in ascus A03-4 and A08-1). The fact that the proportion of SNPs that show Mendelian segregation increase by more strict data filtering strongly indicates that our previous failure to show Mendelian segregation for 32% of the SNPs arose from incomplete filtering of the genome data.

Taken together, the effect of a more stringent SNP detection and detailed inspection of the non-Mendlian SNPs show that the vast majority of the IPO94269 specific SNPs indeed are inherited in a Mendelian manner and that those SNPs that do not show a Mendelian segregation pattern arise from sequencing or assembly errors.

To strengthen the point of Mendelian segregation of SNPs in the *Z. tritici* cross, we have included the following statement in the manuscript:

"To further investigate the occurrence of non-Mendelian SNPs we restricted our analyses to include only SNPs in regions of the genome alignment with high read coverage. […] Further increasing the criteria of SNP detection to SNPs at positions with > 8X read coverage at all occurrences further increased the proportion of SNPs detected in exactly four ascospores to 97% or 98% in ascus A03-4 and A08-1, respectively (Figure 3—figure supplement 2C and 2F, Supplementary file 5)."

We have further updated the Figure 3—figure supplement 2C and F to reflect the effect of different SNP fidelity on the fraction of the SNPs detected in four ascospores.